# Size-Dependent Impact of Magnetic Nanoparticles on Growth and Sporulation of *Aspergillus niger*

**DOI:** 10.3390/molecules27185840

**Published:** 2022-09-09

**Authors:** Zhishang Shi, Yan Zhao, Shuo Liu, Yanting Wang, Qilin Yu

**Affiliations:** 1Key Laboratory of Molecular Microbiology and Technology, Ministry of Education, Department of Microbiology, College of Life Sciences, Nankai University, Tianjin 300071, China; 2Tianjin Key Laboratory of Environmental Remediation and Pollution Control, College of Environmental Science and Engineering, Nankai University, Tianjin 300350, China

**Keywords:** magnetic nanoparticle, biological effect, industrial fungus, cell wall disruption, *Aspergillus niger*

## Abstract

Magnetic nanoparticles (MNPs) are becoming important DNA nanocarriers for genetic engineering of industrial fungi. However, the biological effect of MNPs on industrial fungi remains unknown. In this study, we prepared three kinds of magnetic nanoparticles with different sizes (i.e., 10 nm, 20 nm, and 200 nm) to investigate their impact on the growth and sporulation of the important industrial fungus *Aspergillus niger*. Transmission electron microscopy, X-ray diffraction analysis and Zeta potential analysis revealed that the three kinds of MNPs, including MNP10, MNP20 and MNP200, had uniform size distribution, regular Fe_3_O_4_ X-ray diffraction (XRD) patterns and similar Zeta potentials. Interestingly, although the three kinds of MNPs did not obviously inhibit growth of the fungus, the MNP20 at 500 mg/L strongly attenuated sporulation, leading to a remarkable decrease in spore numbers on culturing plates. Further investigation showed that MNP20 at the high concentration led to drastic chitin accumulation in the cell wall, indicating cell wall disruption of the MNP20-treated fungal cells. Moreover, the MNPs did not cause unusual iron dissolution and reactive oxygen species (ROS) accumulation, and the addition of ferrous ion, ferric ion or the reactive oxygen species scavenger N-acetyl-L-cysteine (NAC) had no impact on the sporulation of the fungus, suggesting that both iron dissolution and ROS accumulation did not contribute to attenuated sporulation by MNP20. This study revealed the size-dependent effect of MNPs on fungal sporulation, which was associated with MNP-induced cell wall disruption.

## 1. Introduction

Filamentous fungi are extensively dispersed in nature and have a tight connection to human life and industry. According to studies, filamentous fungi can produce abundant secondary metabolites such as fungal antagonists, antimicrobials, and anticancer compounds, and the ability of filamentous fungi to produce nanoparticles and antibiofilm polypeptide has also been demonstrated; these substances are widely used in biocontrol, environmental management, and human health [1,2,3,4]. Additionally, filamentous fungi have developed into crucial hosts for the synthesis of organic acids, proteins, and enzymes due to their superior synthesis and secretion abilities [5,6,7]. *Aspergillus niger* (*A. niger*), a representative specie of filamentous fungi and the strain that is GRAS (generally considered as safe) in the food and fermentation industries, has been developed as a cell factory for the synthesis of numerous food enzymes including glucoamylase [8,9,10,11].

Magnetic nanoparticles (MNPs) are magnetic materials with particle size ranging from 1 to 100 nm, and generally composed of metals such as iron, nickel, cobalt and their oxides [12,13,14]. Because of their unique characteristics, including high specific surface area, superparamagnetism, magnetothermal effect, good biocompatibility, and ease of separation, Fe_3_O_4_ nanoparticles are one of the most popular types of MNPs used in biomedical applications, including magnetic hyperthermia, nucleic acid sequencing, targeted drug delivery, magnetic resonance imaging (MRI), tumor cell detection, and magnetofection [13,15,16,17,18,19,20].

Nevertheless, the toxicity of MNPs is a problem that cannot be ignored, and numerous studies have shown that MNPs may be toxic to cells or living organisms [21,22,23]. Nanoparticles can reach lysosomes through the endocytotic pathway and release Fe^2+^, thus generating an excess of reactive oxygen species (ROS) and triggering apoptosis [24]. Similarly, iron oxide nanoparticles produce ROS in the same way, thus causing protein aggregation in neurons, which may damage the central nervous system and may eventually lead to neurodegenerative disease [25]. In addition, MNPs can also cause cell death by damaging organelles after co-incubation of Fe_3_O_4_ nanoparticles with fibroblasts for 48 h, mitochondrial activity decreased [26]. It was also found that high concentrations of Fe_3_O_4_ nanoparticles inhibit *Saccharomyces cerevisiae* growth, as such MNPs can impair mitochondrial function and inhibit ATP synthesis by targeting the respiratory chain complex IV [27].

However, the toxicity of MNPs to filamentous fungi has not been reported. In this study, we investigated the toxicity of Fe_3_O_4_ nanoparticles with different particle sizes and concentrations on *A. niger* from several perspectives, including growth ability, spore-producing ability and cell wall integrity. This study reveals the effect of Fe_3_O_4_ nanoparticles on *A. niger* and its mechanism, which provides a reference for future investigation of the toxicity of MNPs to filamentous fungi and lays a theoretical foundation for the efficient transformation of MNPs for filamentous fungi.

## 2. Results and Discussion

### 2.1. Characterization of the Synthesized MNPs with Different Sizes

To evaluate the effect of MNPs on biological behaviors of the model filamentous fungus *A. niger*, three kinds of MNPs with different sizes were synthesized. A transmission electron microscope (TEM) observation showed that the three MNPs had sizes of about 10 nm for MNP10, 20 nm for MNP20, and 200 nm for MNP200 (Figure 1a). As revealed by X-ray diffraction (XRD) characterization, the three kinds of MNPs shared identical Fe_3_O_4_ crystal XRD patterns, and no unusual peaks were observed (Figure 1b), confirming the high purity of the synthesized Fe_3_O_4_ MNPs. Moreover, dynamic light scattering (DLS) analysis further showed that MNP10, MNP20, and MNP200 had a size distribution similar to the results of TEM (Figure 1c), together with a similar Zeta potential at about 27 mV (Figure 1d). Therefore, the three kinds of MNPs with different sizes but similar surface properties were successfully obtained for further evaluation of the biological effect of MNPs.

### 2.2. Impact of the MNPs on A. niger Growth and Sporulation

While MNPs are becoming one kind of important nanoparticle for genetic manipulation of fungal cells, their effect on fungal growth and sporulation remains unknown. *A. niger*, a model filamentous fungus commonly applied in industrial fermentation, was used to investigate the bioeffect of MNPs. As shown in (Figure 2a, 1 d), the fungus obviously formed filamentous and white colonies on potato dextrose agar (PDA) media containing 500 mg/L of MNPs after one day of culturing. After two days, the fungus rapidly grew to form large colonies with similar diameters on the plates containing different kinds of MNPs (Figure 2a, 2 d). Statistical analysis revealed that the diameters of the fungal colonies in different groups had no significant difference (Appendix A). This indicated that the three kinds of MNPs did not impair the growth of *A. niger* on the PDA medium.

Interestingly, while the control, MNP10 and MNP200 groups had large spore-covering areas in the center, the MNP20 group only had small spore-covering areas (indicated by red and dotted cycles, Figure 2a, 2 d). Statistical analysis confirmed that the diameter (*D*) of spore-covering area in the MNP20 group was significantly lower than that in the other three groups at the MNP concentration of 500 mg/L (Figure 2b). Since the spore-covering area is calculated by π (*D*/2)^2^, the slight difference in diameters will lead to the big difference in the spore-covering areas. Consistently, as revealed by spore counting assays, while 50 mg/L of MNP20 had no obvious impact on spore production, 500 mg/L of MNP20 strongly reduced spore numbers from 16.7 × 10^6^ spores/mL to 8.5 × 10^6^ spores/mL (Figure 2c). In the solid medium, the growth of filamentous fungi originates from the germination of spores. The spores germinate to form tubular hyphae, which extend and branch to form mycelium, and then grow to form a radial colony. After that, part of the aerial mycelium gradually differentiates to form a conidiophore, and then produces asexual spores [28]. Therefore, the above results indicate that MNP20 has no effect on mycelial development, but inhibits spore production. Nevertheless, in contrast, MNP10 at 50 mg/L obviously increased rather than reduced spore numbers to 25.1 × 10^6^ spores/mL, indicating a sporulation-enhancing effect of MNP10 at the low dose (Figure 2c). Taken together, these results indicated that MNP20 strongly impaired the sporulation of *A. niger* at the high dose, while MNP10 enhanced sporulation at the low dose.

### 2.3. Cell Wall Disruption Induced by MNP20

The cell wall, as the surface layer of fungal cells, is the critical structure mediating the interaction between nanoparticles and fungal cells [29,30,31]. Owing to its essential role in fungal sporulation [32], we investigated the impact of the MNPs on cell wall composition. As shown in Figure 3a, the control *A. niger* hyphal cells had a regular tube-like morphology, with chitin stained by Calcofluor White (CFW) equally distributing in the cell wall. Similarly, the hyphal cells treated by MNP10 and MNP200 had normal chitin distribution. In contrast, the hyphal cells treated with MNP20 showed irregularly strong CFW fluorescence throughout the cells (Figure 3a). Fluorescence quantification further confirmed that MNP20 did not change CFW fluorescence intensity at 50 mg/L, but strongly increased the intensity at 500 mg/L (Figure 3b). In addition, aniline blue staining and fluorescence quantification revealed that MNP20 and MNP200 did not change the contents of β-glucan, but MNP10 increased the contents of this component (Figure 3c). Therefore, MNP20 specifically increased the cell wall chitin contents. Since the increase in chitin contents indicates cell wall disruption, these results indicated that MNP20 at the high dose induced cell wall damage in the fungal cells.

### 2.4. Effect of Iron Dissolution from MNPs on Sporulation

Ion dissolution from nanoparticles commonly contributes to the biological effect of nanoparticles [33]. To investigate the mechanism of MNP20-caused sporulation attenuation in *A. niger*, iron dissolution from the three kinds of MNPs was determined. Inductively coupled plasma analysis showed that 50 mg/L of MNP10, MNP20, and MNP200 released 0.23–0.35 mg/L of iron, and 500 mg/L of the nanoparticles released 1.25–1.63 mg/L of iron (Figure 4a). For example, MNP20 released 0.27 mg/L iron at 50 mg/L, and 1.4 mg/L iron at 500 mg/L. In addition, at the high concentration of the MNPs at 500 mg/L, iron dissolution slightly decreased with the increase of the nanoparticle sizes (Figure 4a), which may be attributed to the decreased surface area of the nanoparticles with the increase of sizes.

The effect of dissolved iron from MNP20 on sporulation was then evaluated. While MNP20 at 500 mg/L strongly attenuated spore production, both ferrous ion (Fe^2+^) and ferric ion (Fe^3+^) at the concentration of 0.27 mg/L or 1.4 mg/L had no obvious impact on sporulation. Thus, iron dissolution did not contribute to the inhibitory effect of MNP20 on the sporulation of *A. niger*.

### 2.5. Effect of ROS Accumulation during MNP Treatment on Sporulation

ROS accumulation is another important mechanism of nanotoxicity [34]. ROS levels in the MNP-treated *A. niger* cells were then determined by using DCFH-DA staining. Fluorescence quantification showed that the fungal cells treated by the three kinds of MNPs had ROS levels similar to the control cells (Figure 5a). Moreover, the ROS scavenger NAC could not alleviate the inhibitory effect of MNP20 on sporulation (Figure 5b). Therefore, ROS accumulation was not involved in attenuated sporulation by MNP20.

Based on the above results, we propose a model of MNP20-induced attenuation of sporulation. In the absence of MNPs, the *A. niger* hyphae normally grow and produce asexual spores on conidiophores (Control, Figure 6a). In contrast, under the MNP20 attack, the hyphal cell wall is partially disrupted by the nanoparticles, leading to attenuated sporulation on the conidiophores (Figure 6b). This attenuation is not attributed to both iron dissolution and ROS accumulation, but is to direct cell wall disruption. The possible reason for severe cell disruption caused by MNPs at the size of 20 nm rather than 10 nm or 200 nm may be the optimized interaction between the nanoparticles at the size of 20 nm and the cell wall components. Interestingly, low doses of MNP10 can promote sporulation, which may be due to MNP10 by targeting or promoting the expression of sporulation-related genes, or by altering the external environmental conditions that affect sporulation.

## 3. Materials and Methods

### 3.1. Materials

Ferrous chloride, ferric chloride and ammonium hydroxide were purchased from Sigma (St. Louis, MO, USA). All reagents were used without further purification.

PDA solid medium contains 20% potato extraction, 2% agar and 2% glucose. After autoclaving and cooling to 50–60 °C (when the medium is about to solidify), MNPs of specific particle size and concentration are added under aseptic conditions, mixed sufficiently to evenly disperse the MNPs into the medium, and poured into petri dishes of 10 cm diameter. PDA liquid medium contains 20% potato extraction and 2% glucose. After autoclaving, it is dispensed into sterile test tubes under aseptic conditions.

### 3.2. Methods

#### 3.2.1. Strains and Growth Conditions

The toxicity of Fe_3_O_4_ nanoparticles was evaluated using *A. niger* wild-type (WT) strains. *A. niger* was inoculated in PDA solid medium and incubated in 28 °C to obtain spores and in PDA liquid medium under a shaking table at 30 °C to obtain mycelia. Incubation with shaking not only promotes oxygen dissolution in the liquid, but also avoids the aggregation and precipitation of MNPs.

#### 3.2.2. Preparation and Characterization of Fe_3_O_4_ Nanoparticles with Different Particle Sizes

To synthesize the MNP10 (10 nm) and MNP20 (20 nm), 0.4 g of ferrous chloride and 1.1 g of ferric chloride were dissolved in distilled water by magnetic stirring under 80 °C. 5 mL of ammonium hydroxide were then slowly added into the solution. After further stirring for 2 h, the MNPs were centrifuged at 5000 rpm for 5 min, obtaining MNP20. The supernatant was further centrifuged at 12,000 rpm for 10 min, obtaining MNP10. The nanoparticles were washed by distilled water five times and dried by a vacuum freezing drier for further use. The nanoparticles MNP200 (200 nm) were synthesized by using the hydrothermal method according to previous reports [35].

The morphology of MNP10, MNP20 and MNP200 was characterized by TEM (Tecnai G2F-20, FEI, Hillsboro, OR, USA). Size distribution and Zeta potentials of the nanoparticles were detected by a DLS (Malvern Panalytical, Zetasizer Nano ZS0303081003). The X-ray diffraction spectra of the nanoparticles were characterized by a XRD analyzer (SmartLab-SE, Rigaku, Tokyo, Japan).

#### 3.2.3. Growth Inhibition Assays and Measurement of Spore Production Capacity

In order to evaluate the effect of different particle sizes and concentrations of Fe_3_O_4_ nanoparticles on the growth of *A. niger*, PDA solid media containing three particle sizes of Fe_3_O_4_ nanoparticles at 50 mg/L and 500 mg/L were prepared in advance. A small number of spores were picked from the PDA solid medium containing the WT strain of *A. niger* and planted in the center of the PDA plates containing different particle sizes and concentrations of Fe_3_O_4_ nanoparticles, and the dishes were incubated at 28 °C. Photographs were taken every 24 h and colony diameters were measured with a ruler. After six days of growth, the spores on the plates were washed off with double distilled water (ddH_2_O) and suspended in 10 mL ddH_2_O, shaken for 20–30 min to make spore suspensions, and the suspension was sampled and the spores were counted by a hemocytometer to determine the effect of Fe_3_O_4_ nanoparticles on the sporulation capacity of *A. niger*.

#### 3.2.4. Cell Wall Staining

CFW is a fluorescent stain for fungal cell wall staining and chitin content analysis, since it can bind fungal cell wall chitin and absorb ultraviolet light to make mycelium and spores emit bright fluorescence [36]. To observe whether Fe_3_O_4_ nanoparticles caused cell wall damage, mycelia were cultured and collected by the above method, washed twice with PBS and suspended in 1 mL PBS, then stained with CFW (Sigma, St. Louis, MO, USA) at a final concentration of 0.5 μg/mL for 10 min, and the mycelia were washed three times with PBS. The cell wall morphology was observed by fluorescence microscopy (BX-41, Olympus, Tokyo, Japan) with the blue filter set.

#### 3.2.5. Chitin Measurement

Chitin is an essential component of the fungal cell wall, and the content of this component increases when the cell wall is damaged. Therefore, cell wall chitin can reflect the degree of cell wall damage. To detect the chitin content in the mycelial cell wall, CFW staining was performed as described above followed by three washes with PBS, and the supernatant was added to 96-well microplates and the fluorescence density (FLU) (excitation wave 325 nm, emission wave 435 nm) was measured by a fluorescence microplate reader (Enspire, Perkinelmer LLC, Waltham, MA, USA). Afterwards, the excess supernatant was discarded, the mycelia was dried and the dry weight was weighed, and the relative fluorescence intensity was calculated as FLU divided by the dry weight of the mycelia.

#### 3.2.6. β-1,3-Glucan Assay

Mycelia cultured in PDA liquid medium were collected, washed once with TE solution (10 mM Tris and 1 mM EDTA, pH 8.0), and resuspended in 500 μL TE solution, added with final concentration of 1 M NaOH, mixed well, and incubated at 80 °C for 30 min. 2 mL of AB mixture was added (0.03% aniline blue, 3.678% glycine, 1.5% (*v*/*v*) concentrated hydrochloric acid, pH 9.5, store at 4 °C in the dark), and they were incubated at 50 °C for 30 min and then placed at room temperature for 30 min. The FLU of the cells was measured (excitation wave 400 nm, emission wave 460 nm). The excess supernatant was discarded. The mycelia were dried and the dry weight of the mycelia was weighed. The relative fluorescence intensity was calculated as FLU divided by the dry weight of the mycelia.

#### 3.2.7. Statistical Analysis

The experiments were performed with three replicates (*n* = 3). The data were shown with the means and the standard errors. The significance of difference between the groups was analyzed by a student’s *t*-test method (*p* < 0.05) using the Statistical Product and Service Solutions software (IBM, Armonk, NY, USA).

## 4. Conclusions

In conclusion, this study investigated the effect of MNPs at different sizes on growth and sporulation of the model industrial fungus *A. niger*. The MNPs at three sizes, i.e., 10 nm, 20 nm, and 200 nm, had no obvious impact on the growth of the fungus on normal PDA plates. Interestingly, the MNPs at the size of 20 nm strongly attenuated the sporulation of the fungus, leading to a remarkable decrease in spore numbers on the PDA plates. Further investigation revealed that the attenuation of sporulation caused by MNP20 is attributed to cell wall disruption rather than iron dissolution and ROS accumulation. The mechanism remains to be further discussed, and it has been shown that in *Aspergillus nidulans*, deletion of the *MtlA* gene, which is closely related to the cell wall integrity signaling pathway, can lead to the reduction of conidia formation [37]. Therefore, in future research, we can explore the changes of sporulation-related genes under the induction conditions of MNP20 or other cell wall-damaging agents such as snailase so as to further explore the mechanism of toxicity of MNPs to *A. niger*. In addition, magnetofection, which uses MNPs to deliver nucleic acids into cells under the action of a magnetic field, has been recognized as an efficient and ideal transformation method [38]. However, there are still few related studies, and there is no report on the gene delivery of MNPs to filamentous fungi. Therefore, this study sheds novel light on size-dependent effect of MNPs on fungal cell behaviors, which will guide choose of MNPs during their application in genetic manipulation.

## Figures and Tables

**Figure 1 molecules-27-05840-f001:**
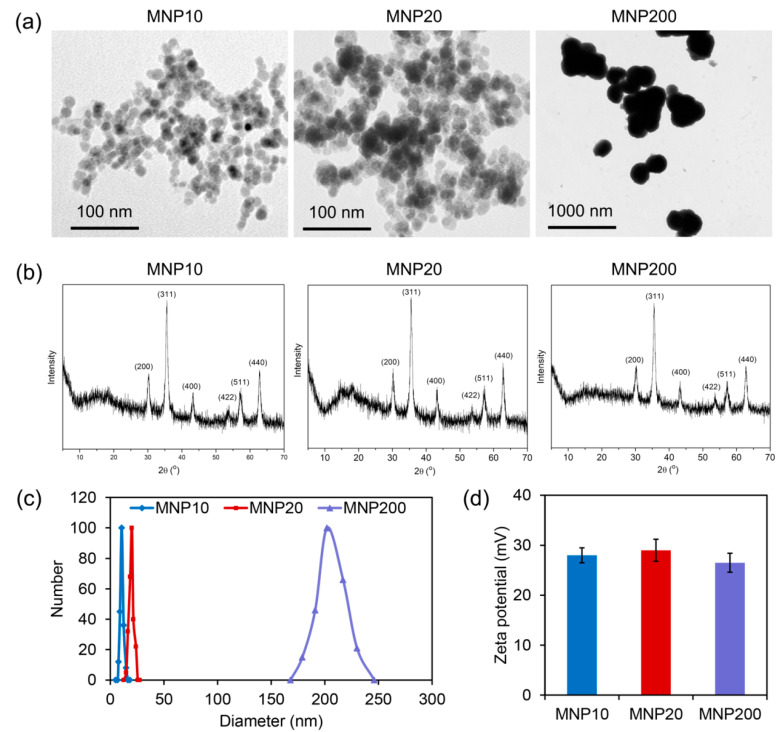
Characterization of the synthetic MNPs with different sizes. (**a**) TEM images of the synthetized nanoparticles MNP10 (size ≈ 10 nm), MNP20 (size ≈ 20 nm), and MNP200 (size ≈ 200 nm). (**b**) XRD patterns of the MNPs. (**c**) Size distribution of the MNPs revealed by DLS analyzer. (**d**) Zeta potentials of the MNPs.

**Figure 2 molecules-27-05840-f002:**
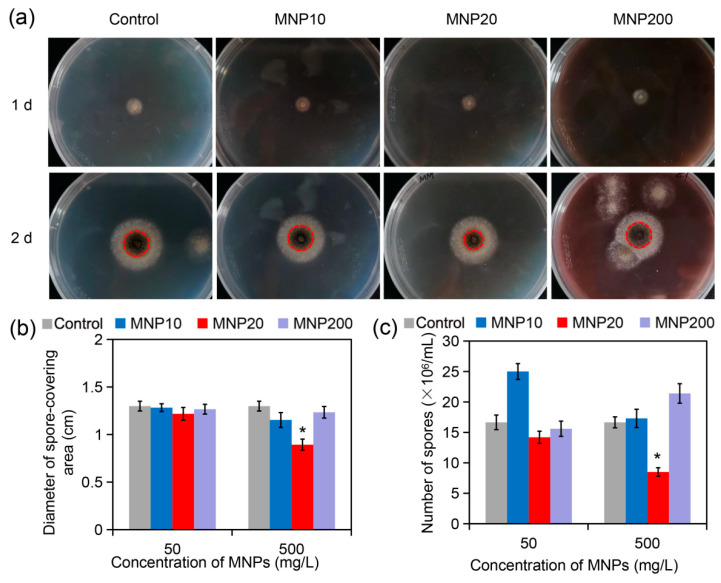
Impact of the MNPs with different sizes on growth and sporulation of *A. niger*. (**a**) Images of the *A. niger*-culturing PDA plates containing the MNPs at 500 mg/L. The plates were cultured at 30 °C for 1 d (24 h) or 2 d (48 h), followed by photographing. The red cycles indicated the areas of spore covering. (**b**) Diameters of spore-covering areas in different groups after two days of culturing. (**c**) Number of spores in different groups. The spores in each plate after six days of culturing were washed by 10 mL of distilled water, and the spores were counted by using hemocytometers. The asterisks (*) indicate significant difference between the MNP20 group and other groups (*p* < 0.05).

**Figure 3 molecules-27-05840-f003:**
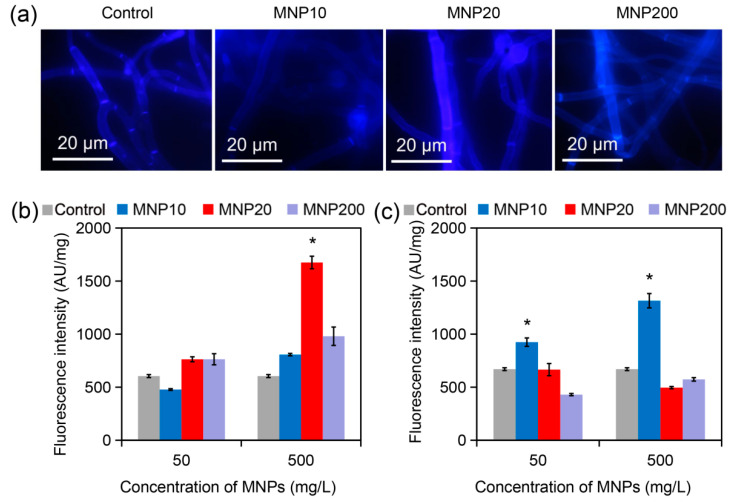
Observation and analysis of cell wall components in *A. niger* treated by the MNPs for two days. (**a**) Fluorescence microscope images of the *A. niger* hyphae. The hyphae were treated by the MNPs (500 mg/L) for two days, harvested and stained by CFW for observation of chitin distribution. (**b**) Cell wall chitin contents of the treated hyphae revealed by CFW staining and fluorescence quantification. (**c**) β-glucan contents of the treated hyphae revealed by aniline blue staining and fluorescence quantification. The asterisks (*) indicate significant difference between the MNP20 group and other groups (*p* < 0.05).

**Figure 4 molecules-27-05840-f004:**
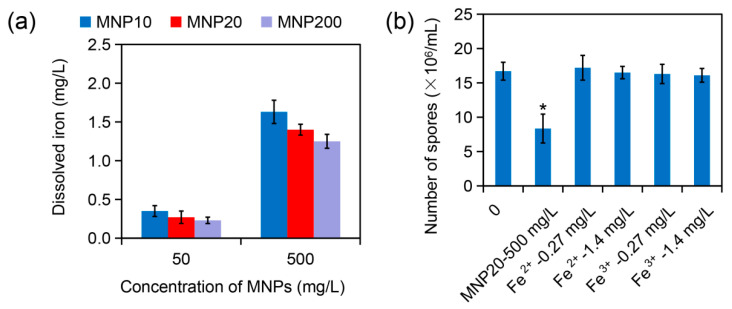
Iron dissolution from the MNPs and its effect on sporulation *A. niger* under MNP treatment. (**a**) Iron dissolution from the MNPs at the concentration of 50 mg/L or 500 mg/L in liquid PDA medium. (**b**) Effect of ferrous ion (Fe^2+^) and ferric ion (Fe^3+^) on sporulation of *A. niger*. The asterisks (*) indicate a significant difference between the MNP20 group and other groups (*p* < 0.05).

**Figure 5 molecules-27-05840-f005:**
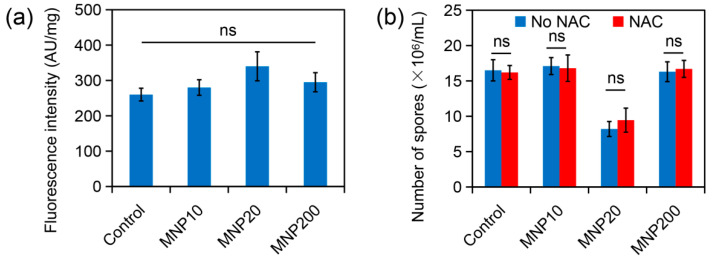
ROS accumulation and its contribution on sporulation of *A. niger* under MNP treatment. (**a**) ROS levels in MNP-treated *A. niger* cells revealed by DCFH-DA staining and fluorescence quantification. (**b**) Effect of the ROS scavenger NAC on sporulation of *A. niger*. Note that the addition of NAC has no obvious impact on sporulation. The letters “ns” indicate no significant difference between the groups (*p* < 0.05).

**Figure 6 molecules-27-05840-f006:**
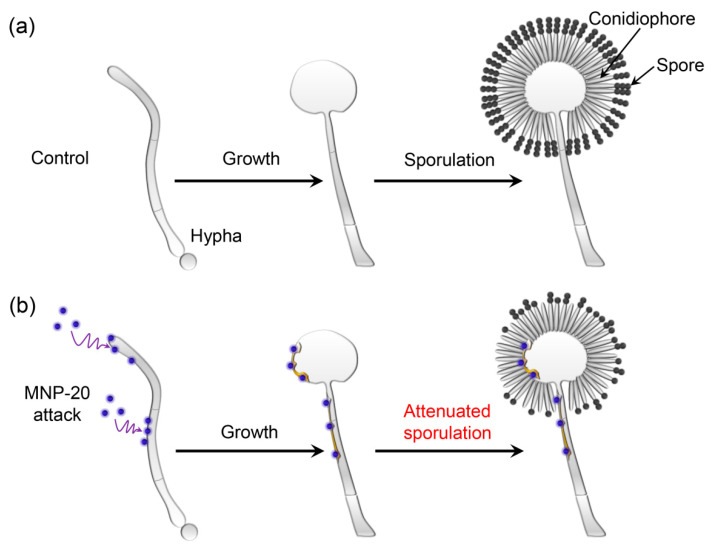
Schematic illustration of attenuated sporulation of *A. niger* induced by MNP20. (**a**) Normal growth and sporulation of the *A. niger* hyphae on the conidiophores in the control group without MNP treatment. (**b**) Normal growth but attenuated sporulation in the MNP20-treated hypha. The attack of MNP20 led to partial cell wall damage, leading to attenuated sporulation on the conidiophores.

## Data Availability

Not applicable.

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
