# Peer review of "Size-Dependent Impact of Magnetic Nanoparticles on Growth and Sporulation of Aspergillus niger"

_molecules, 2022, doi:10.3390/molecules27185840_

Round 1

Reviewer 1 Report

The authors reported on the effect of Fe3O4 nano-particle (NPs) sizes and  concentrations in the growth and sporulation of Aspergillus niger. My comments are below:

- The authors should detail the PDA solid preparation procedure.

- The distribution of Fe3O4 NPs into PDA wasn't mentioned at the beginning and the end of the tests of fungal sporulation. Fe3O4 NPs can agglomerate to become bigger particles during the PDA preparation and fungal sporulation, that may change the results of Fe3O4 size influences.

- The authors should explain why the fluorescence intensity of MNP20  at 500 mg/L is significantly higher than that of other. And why the number of spores is so high with MNP10 at 50 mg/L?

- The authors said no obvious difference in iron release between the three kinds of MNPs, but the concentration of Fe ion decreased with the increase of NPs sizes. 

- Some typing words should be corrected.

- Some abbreviated word should be classified at the first mention.

Author Response

Response to Reviewer 1 Comments

We appreciate your valuable comments. Please find our point-by-point responses to your comments below. We use the following color codes to make it easier to see the comment, the response, and the changes made in manuscript:

Reviewer Comments:              blue

Authors’ Response:          black

Manuscript changes:         red (“Track Changes” function in MS Word)

  1. The authors should detail the PDA solid preparation procedure.

Reply: We have added the description of PDA solid preparation procedure in 2.1. Materials. Please see lines 74-80 of the article for more details on this.

  1. The distribution of Fe3O4 NPs into PDA wasn't mentioned at the beginning and the end of the tests of fungal sporulation. Fe3O4 NPs can agglomerate to become bigger particles during the PDA preparation and fungal sporulation, that may change the results of Fe3O4 size influences.

Reply: We have added the procedures of PDA solid and liquid mediums preparation, and illustrated that Fe3O4 NPs do not aggregate and precipitate during the experiment. Please see lines 77 and 88 of the article for more details on this.

  1. The authors should explain why the fluorescence intensity of MNP20 at 500 mg/L is significantly higher than that of other. And why the number of spores is so high with MNP10 at 50 mg/L?

Reply: Concerning the fluorescence intensity, we have added the mechanism of Calcofluor White (CFW) staining and the relationship between cell wall damage and chitin content, please see lines 117-119 and 126-128 of the article for more details on this. And we propose a possible reason for the low concentration of MNP10 to promote spore formation in lines 276-279 of the article.

  1. The authors said no obvious difference in iron release between the three kinds of MNPs, but the concentration of Fe ion decreased with the increase of NPs sizes.

Reply: We have added the description of the slight decrease in iron release between the three kinds of nanoparticles. Please see Page 7.

  1. Some typing words should be corrected.

Reply: The typing errors were checked and revised. All changes to typing words have been marked in red in the article.

  1. Some abbreviated word should be classified at the first mention.

Reply: The full names of abbreviated words have been supplied. All changes to classified abbreviated word have been marked in red in the article.

Reviewer 2 Report

In this wrok, the author investigated the effect of MNPs at different different sizes on growth 265 and sporulation of the fungus A. niger. The author futher studied the attenuation of sporulation caused by MNP20 and proposed a cell wall disruption hypothesis. It is overall a sound work, but many concerns need to be solved. 

1. The author should clarity the the reason for using MNP in fungus A. niger to rationalize the study. If none use MNP in fungus A. niger, this work will be much less attractive for readers. 

2. Compare Figure 2b and c, MNP should little impact on the diameter of psore-covering but strong effect on the number of spores. Could the author explain this inconsistence? 

3. A sound cell wall disruption hypothesis was proposed by the author to explain the the attenuation of sporulation caused by MNP20. Experimental supports, if possible, will strengthen this hypothesis. 

Author Response

Response to Reviewer 2 Comments

We appreciate your valuable comments. Please find our point-by-point responses to your comments below. We use the following color codes to make it easier to see the comment, the response, and the changes made in manuscript:

Reviewer Comments:              blue

Authors’ Response:          black

Manuscript changes:         red (“Track Changes” function in MS Word)

  1. The author should clarity the the reason for using MNP in fungus A. niger to rationalize the study. If none use MNP in fungus A. niger, this work will be much less attractive for readers.

Reply: We have further elaborated on the significance of MNPs for Aspergillus niger. Please see lines 301-307 of the article for more details on this:

“In addition, magnetofection, which uses MNPs to deliver nucleic acids into cells under the action of a magnetic field, has been recognized as an efficient and ideal transformation method [38]. However, there are still few related studies, and there is no report on the gene delivery of MNPs to filamentous fungi.”

  1. Compare Figure 2b and c, MNP should little impact on the diameter of spore-covering but strong effect on the number of spores. Could the author explain this inconsistence?

Reply: We have added the development process of filamentous fungi in solid medium in lines 187-192 of the article, which in turn explains the cause of this phenomenon:

“Since the spore-covering area is calculated by π·(D/2)2, the slight difference in diameters will lead to the big difference in the spore-covering areas.”

  1. A sound cell wall disruption hypothesis was proposed by the author to explain the the attenuation of sporulation caused by MNP20. Experimental supports, if possible, will strengthen this hypothesis.

Reply: We have added a possible explanation for this hypothesis and an experimental protocol to test this explanation in lines 294-300 of the article:

“Certainly, the mechanism remains to be further discussed, and it has been shown that in Aspergillus nidulans, deletion of the MtlA gene, which is closely related to the cell wall integrity (CWI) signaling pathway, can lead to reduction of conidia formation [37]. Therefore, in future research, we can explore the changes of sporulation-related genes under the induction conditions of MNP20 or other cell wall-damaging agents such as snailase, so as to further explore the mechanism of toxicity of MNPs to A. niger.”